# GENERALIST SCANNER MEETS SPECIALIST LOCATOR: A SYNERGISTIC COARSE-TO-FINE FRAMEWORK FOR ROBUST GUI GROUNDING

## ABSTRACT

Grounding natural language queries in graphical user interfaces (GUIs) presents a challenging task that requires models to comprehend diverse UI elements across various applications and systems, while also accurately predicting the spatial coordinates for the intended operation. To tackle this problem, we propose *GMS: Generalist Scanner Meets Specialist Locator*, a synergistic coarse-to-fine framework that effectively improves GUI grounding performance. *GMS* leverages the complementary strengths of general vision-language models (VLMs) and small, task-specific GUI grounding models by assigning them distinct roles within the framework. Specifically, the general VLM acts as a *'Scanner'* to identify potential regions of interest, while the fine-tuned grounding model serves as a *'Locator'* that outputs precise coordinates within these regions. This design is inspired by how humans perform GUI grounding, where the eyes scan the interface and the brain focuses on interpretation and localization. Our whole framework consists of five stages and incorporates hierarchical search with cross-modal communication to achieve promising prediction results. Experimental results on the ScreenSpot-Pro dataset show that while the *'Scanner'* and *'Locator'* models achieve only $2.0\%$ and $3.7\%$ accuracy respectively when used independently, their integration within *GMS* framework yields an overall accuracy of $35.7\%$, representing a $10\times$ improvement. Additionally, *GMS* significantly outperforms other strong baselines under various settings, demonstrating its robustness and potential for general-purpose GUI grounding.

## 1 INTRODUCTION

Grounding natural language queries in graphical user interfaces (GUIs) requires models to predict accurate coordinates for user-specified actions, enabling applications in agent control, device automation, and accessibility (Wang et al., 2025a; Nguyen et al., 2025; Zhang et al., 2025a; Tang et al., 2025b). As vision-language models (VLMs) advance in multimodal reasoning, GUI grounding emerges as a key benchmark for evaluating their interactive capabilities (Hui et al., 2025; Wang et al., 2025b; Li et al., 2025; Cheng et al., 2024; Liu et al., 2024). GUI grounding is challenging due to the diverse structures, styles, and semantics of interfaces across platforms. It requires fine-grained understanding of both textual and non-textual elements, dense visual layouts, and context-dependent functions, making accurate interpretation difficult (Li et al., 2025; Wu & Xie, 2024; Wu et al., 2025a).

Existing approaches can be broadly categorized into two groups: (i) Training-based methods either fine-tune base vision-language models, such as Qwen2-VL-7B, to directly predict grounding coordinates, or employ reinforcement learning techniques, such as GRPO, to induce multi-step reasoning processes that ultimately localize the target region (Gou et al., 2025; Wu et al., 2024; Shao et al., 2024). Although fine-tuning improves task-specific performance, it often sacrifices the model's capacity for self-correction and adaptive reasoning. Reinforcement learning methods offer greater flexibility and generalization, but they incur substantial computational overhead and suffer from slow inference due to the complexity of the reasoning procedures they require (Luo et al., 2025a; Zhou et al., 2025; Lu et al., 2025; Liu et al., 2025; Tang et al., 2025a). (ii) Training-free methods seek to bypass the cost of retraining by leveraging pre-trained models. These include recursive zoom-in techniques that iteratively refine grounding predictions and planner-based strategies that

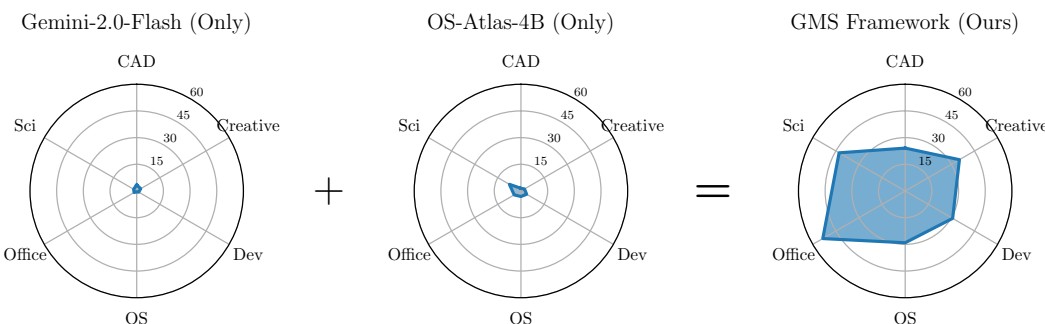

Figure 1: The two original models individually perform poorly on the GUI grounding task, with average accuracies below 4%. Under our GMS framework, where each model specializes in its strengths and collaborates effectively, the overall accuracy reaches 36%, which is nearly 10× higher than their standalone performance.

utilize general models to guide localizers (Li et al., 2025; Wu et al., 2025a; Nguyen, 2025; Luo et al., 2025b; Ge et al., 2025; Zhang et al., 2024; Wang et al., 2024a). However, zoom-in strategies are highly sensitive to initial prediction errors and lack any verification mechanism, making them fragile in practice. Planner-based approaches mitigate this to some extent by introducing coordination between models, but they continue to rely on general models to produce bounding boxes. Since these general models are not trained explicitly for precise localization, the resulting predictions are often inaccurate, and errors tend to propagate throughout the grounding process.

To address these limitations, we proposes **GMS**: **G**eneralist **S**canner **M**eets **S**pecialist Locator, a synergistic coarse-to-fine framework. GMS integrates the complementary strengths of general-purpose and task-specific models to construct a training-free, modular grounding pipeline, as shown in Figure 2. The design of GMS is inspired by the human visual cognition process, in which broad perceptual scanning is followed by focused attention for fine-grained decision making. Accordingly, the general-purpose vision-language model operates as a 'Scanner' that identifies high-confidence candidate regions at a coarse level, while a fine-tuned GUI grounding model functions as a specialist 'Locator' that predicts precise coordinates within the selected regions. The GMS framework consists of five modules that enable coarse-to-fine localization: (1) *Hierarchical attention allocation*, where the "Scanner" partitions the screen into coarse grids and selects semantically relevant regions; (2) *Iterative focus refinement*, where ambiguous areas are recursively zoomed in through semantically guided subdivision; (3) *Cross-modal verification*, where the "Locator" proposes coordinates that are validated by the "Scanner" to suppress false positives; (4) *Multi-agent consensus*, where the "Scanner" and "Locator" predictions are fused with asymmetric weighting for robust agreement; and (5) *Adaptive resolution enhancement*, where multi-scale late fusion reconciles coarse semantic cues with fine pixel-level localization. This design creates a cognitively inspired perception and action loop, outperforming prior pipelines in both robustness and precision.

By addressing key limitations in existing approaches, such as the absence of verification mechanisms in zoom-in methods and the imprecise localization capabilities in planner-based strategies, GMS demonstrates strong generalization and efficiency. Empirical results on the benchmark dataset confirm that our proposed framework achieves substantial performance gains. Notably, even when initialized with two individually weak models, GMS improves grounding accuracy from below 4% to 36%, as illustrated in Figure 1. In summary, our contributions are threefold:

(1) We introduce *GMS*, a training-free multi-agent framework that emulates human-like grounding by assigning complementary roles to generalist and specialist models, achieving substantial gains without additional fine-tuning.

(2) Experiments on the ScreenSpot-Pro dataset show that *GMS* improves performance by more than 10× with weak model pairs and consistently outperforms other strong baselines, demonstrating both robustness and generalizability.

(3) We conduct extensive evaluations, including test-time scaling and ablations, to validate the framework. The results show that agents are most effective when specialized, leading to robust performance across diverse and challenging scenarios.

Figure 2: A simplified illustration captures the motivation and design of the proposed GMS framework. General-purpose VLMs exhibit broad visual and semantic understanding but often fail to produce accurate coordinate predictions. In contrast, grounding fine-tuned VLMs offer precise localization capabilities but lack the high-level reasoning required for complex tasks. Individually, both models tend to produce incorrect outputs. However, by leveraging their complementary strengths and assigning the general VLM as a 'Scanner' and the grounding VLM as a 'Locator', mimicking the roles of human eyes and brain, this system can effectively generate correct answers.

## 2 RELATED WORKS

Recent advances in vision-language models (VLMs) have significantly improved multimodal understanding by jointly learning visual and textual representations (Comanici et al., 2025; OpenAI et al., 2024; Anthropic, 2025; OpenAI, 2025; Bai et al., 2025). Building on their strong capabilities, recent work investigates more interactive and grounded scenarios, where models must not only interpret visual content but also localize and manipulate elements within images. This direction naturally extends to GUI grounding, which maps user instructions to actionable interface elements in GUIs.

In parallel, recent years have seen significant advances in research on GUI agents, evolving from rule-based web automation to general-purpose interface control across platforms such as mobile and desktop. A persistent challenge is the reliable localization of interface elements, which remains a key bottleneck for robust automation (Nakano et al., 2022; Zhang et al., 2025b; Wang et al., 2024a). The emergence of vision-language models marks a shift toward perception-driven grounding by leveraging both visual and textual inputs, without depending solely on structured metadata. Recent work improves robustness by fine-tuning VLMs on GUI-specific datasets, resulting in models that predict element coordinates with higher precision (Gou et al., 2025; Wu et al., 2024; Gu et al., 2025; Qin et al., 2025). Some studies extend this direction using reinforcement learning approaches for multi-step decision-making with interpretable intermediate outputs (Tang et al., 2025c; Luo et al., 2025a; Wu et al., 2025b). In parallel, training-free approaches explore dual-system models, iterative zoom-in mechanisms, and the repurposing of general purpose models as planners to guide action selection (Wu et al., 2025a; Li et al., 2025). However, existing methods often overlook collaborative agent architectures, in which two specialized models assume distinct roles aligned with their respective strengths. Such cooperation presents a promising direction for integrating complementary model capabilities in GUI grounding.

## 3 METHODOLOGY

GUI grounding poses a dual challenge: it requires both global semantic understanding and precise spatial localization. Prior approaches often rely on a single model to handle both tasks simultaneously, leading to trade-offs that limit overall performance. Inspired by the dual-stream hypothesis in visual cognition, which separates the "what/where" pathway from the "how" pathway in human perception, we propose **GMS**: **G**eneralist Scanner **M**eets **S**pecialist Locator, a framework that explicitly decomposes the grounding task into two specialized agents: a generalist vision-language model acting as a 'Scanner', and a fine-tuned GUI grounding model serving as a 'Locator'. GMS follows a coarse-to-fine strategy across five stages, with the detailed process illustrated in Figure 3.

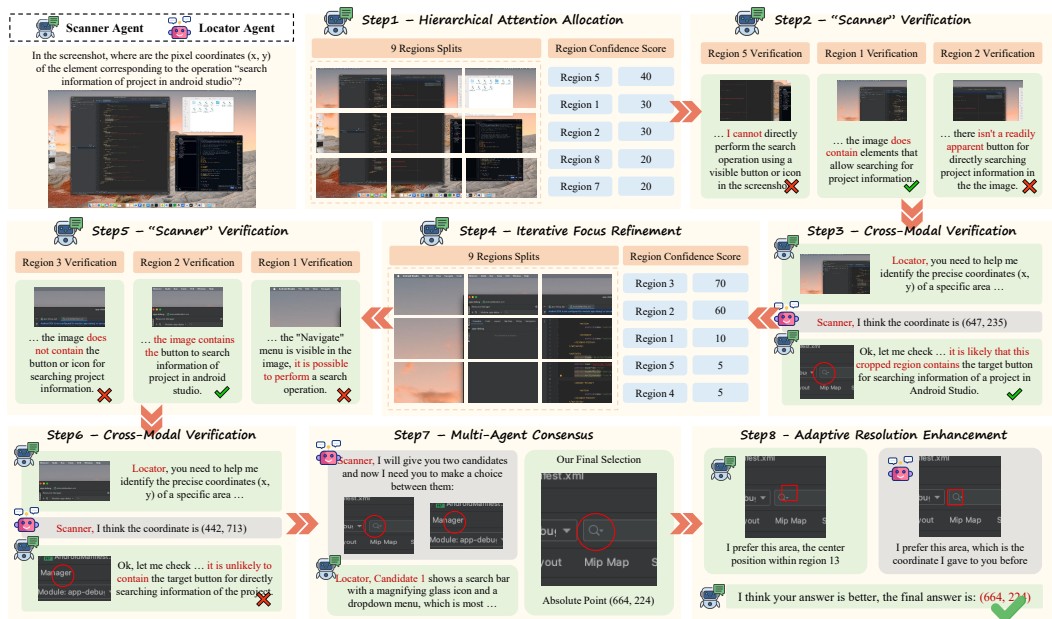

Figure 3: A detailed illustration of the proposed GMS framework highlights its multi-stage and hierarchical process. The 'Scanner' module mimics human vision by constraining the search space and identifying regions of interest, while the 'Locator' module emulates cognitive decision-making to determine precise coordinates. Each agent performs its dedicated role, yet they cooperate seamlessly within the framework, leveraging their complementary strengths to accurately predict the target coordinates.

Formally, let the GUI screen be an image $I \in \mathbb{R}^{H \times W \times 3}$ and a natural language instruction as $Q$. The goal is to predict a pixel coordinate $p = (x^*, y^*) \in [0, W] \times [0, H]$ corresponding to the GUI element described in $Q$. GMS achieves this through the following stages:

## 3.1 HIERARCHICAL ATTENTION ALLOCATION

Human visual attention operates in a coarse-to-fine manner, allocating cognitive resources hierarchically across the scene. Psychological studies show that within the first 300ms of exposure, humans can perform scene parsing to identify regions of interest prior to fine-scale analysis. GMS emulates this behavior via adaptive grid partitioning and region-level semantic scoring.

Specifically, we begin by decomposing the screen into a $3 \times 3$ grid:

$$R = \{R_1, R_2, \ldots, R_9\}, \quad R_i \subset I.$$

Each region $R_i$ is defined by its bounding box $B_i = [x_1^i, y_1^i, x_2^i, y_2^i]$. The generalist vision-language model (e.g., GPT, Gemini) then acts as the 'Scanner', which evaluates each region's semantic relevance to the query $Q$ by computing:

$$s_i = \texttt{Select}(Inst_{\text{selection}}, Q, R_i), \quad s_i \in [0, 100].$$

The top-$k$ scoring regions are selected to form the candidate set $R_{\text{top}}$. Note that our choice of $3 \times 3$ reflects a trade-off between semantic coverage and token cost. While finer grids (e.g., $4 \times 4$) increase resolution, they incur diminishing returns in early-stage filtering and increase computational load.

## 3.2 ITERATIVE FOCUS REFINEMENT

One-shot attention allocation often fails in high-density GUI scenes due to:

- Semantic ambiguity from visually similar but functionally distinct elements.
- Contextual dependencies requiring reasoning over inter-element relations (e.g., "checkbox next to the password field").

To mitigate this, we design a recursive depth-first search (DFS) refinement process. At each level $l$, regions $R^{(l)}$ are recursively subdivided into $3 \times 3$ subgrids. The 'Scanner' re-applies the selection function:

$$R^{(l+1)} = \texttt{Select}(R^{(l)}, Inst_{\text{selection}}, Q),$$

until one of two stopping conditions is met: (i) the region's width or height is below a threshold (e.g., $< 600$ px), or (ii) subsequent verification (section 3.3) indicates insufficient confidence. Unlike naïve zoom-in approaches, each refinement step is semantically informed and is grounded in a verification loop to suppress false positives.

## 3.3 CROSS-MODAL VERIFICATION

While generalist models excel at region-level semantic matching, they often suffer from false confidence due to hallucinations or overgeneralization. To correct this, we introduce a cross-modal verification mechanism that uses the specialist 'Locator' as a factuality check.

For each selected region $R^{(l)}$, the 'Locator' agent predicts a coordinate:

$$\hat{p}_l = \texttt{GUIGround}(Q, R^{(l)}).$$

A crop $C_l$ of size $125 \times 125$ pixels is extracted around $\hat{p}_l$, providing localized context. The 'Scanner' agent then performs verification:

$$v_l = \texttt{Verify}(C_l, Q, Inst_{\text{verification}}), \quad v_l \in \{0, 1\}.$$

The patch size is carefully chosen to balance context and specificity. Empirically, $125 \times 125$ provides sufficient local cues while avoiding dilution from too many unrelated UI elements.

## 3.4 MULTI-AGENT CONSENSUS

After multiple rounds of verification, we could obtain $t$ candidate crops:

$$\mathcal{C} = \{(C_1, v_1), \ldots, (C_t, v_t)\}, \quad v_l \in \{0, 1\}.$$

Selecting the best candidate is framed as a multi-agent consensus problem. Instead of naïve majority voting, we adopt an asymmetric weighting strategy, reflecting each agent's relative expertise:

(i) The 'Scanner' agent contributes global context understanding and high-level semantic reasoning across multiple candidate regions.

(ii) The 'Locator' agent contributes fine-grained spatial precision and reliable confidence estimation within localized regions.

The 'Scanner' agent is instructed as follows:

$$\hat{i} \leftarrow \texttt{Eval}(Inst_{\text{evaluation}}, \mathcal{C}), \quad C^* = C_{\hat{i}}.$$

This step ensures that the selected region maximally aligns with both semantic and spatial constraints of the instruction $Q$.

## 3.5 ADAPTIVE RESOLUTION ENHANCEMENT

The final prediction requires resolving discrepancies between coarse attention and fine spatial cues. Generalist vision-language models operate on low-resolution patches, while the specialist operates on raw pixels. To bridge this, we design a multi-scale late fusion module.

First, we upscale $C^*$ by $\times 5$ in both dimensions to improve resolution. A $5 \times 5$ grid is imposed, followed by a $3 \times 3$ subgrid within the selected region. The 'Scanner' estimates a coarse point:

$$p_{\text{scanner}} = \texttt{Center}(z^*).$$

In parallel, the 'Locator' provides a direct prediction:

$$p_{\text{locator}} = \texttt{GUIGround}(Q, C^*).$$

The final decision is delegated to the stronger 'Scanner' agent:

$$p_{\text{final}} = \texttt{Decide}(Q, C^*, \{p_{\text{scanner}}, p_{\text{locator}}\}, Inst_{\text{decision}}).$$

This fusion mechanism leverages multi-resolution reasoning, ensuring that the final coordinate prediction is both semantically coherent and spatially precise. The design echoes principles from receptive field theory, where layered attention enhances perceptual granularity.

| Base Model | Development | | | Creative | | | CAD | | | Scientific | | | Office | | | OS | | | Average | | |
|---|---|---|---|---|---|---|---|---|---|---|---|---|---|---|---|---|---|---|---|---|---|
| | text | icon | avg | text | icon | avg | text | icon | avg | text | icon | avg | text | icon | avg | text | icon | avg | text | icon | avg |
| GPT-4o | 1.3 | 0.0 | 0.7 | 1.0 | 0.0 | 0.6 | 2.0 | 0.0 | 1.5 | 2.1 | 0.0 | 1.2 | 1.1 | 0.0 | 0.6 | 0.0 | 0.0 | 0.0 | 1.3 | 0.0 | 0.8 |
| Gemini-2.0-Flash | 0.6 | 2.1 | 1.3 | 4.5 | 0.0 | 2.6 | 3.6 | 3.1 | 3.4 | 2.1 | 1.8 | 2.0 | 2.3 | 0.0 | 1.7 | 0.0 | 1.1 | 0.5 | 2.5 | 1.3 | 2.0 |
| Gemini-2.5-Flash-Lite | 1.3 | 0.0 | 0.7 | 2.0 | 4.2 | 2.9 | 7.6 | 1.6 | 6.1 | 4.2 | 0.9 | 2.8 | 2.3 | 3.8 | 2.6 | 0.0 | 1.1 | 0.5 | 3.2 | 1.8 | 2.7 |
| CogAgent-18B | 14.9 | 0.7 | 8.0 | 9.6 | 0.0 | 5.6 | 7.1 | 3.1 | 6.1 | 22.2 | 1.8 | 13.4 | 13.0 | 0.0 | 6.5 | 5.6 | 0.0 | 3.1 | 12.0 | 0.8 | 7.7 |
| Aria-UI | 16.2 | 0.0 | 8.4 | 23.7 | 2.1 | 14.7 | 7.6 | 1.6 | 6.1 | 27.1 | 6.4 | 18.1 | 20.3 | 1.9 | 16.1 | 4.7 | 0.0 | 2.6 | 17.1 | 2.0 | 11.3 |
| Claude (Computer Use) | 22.0 | 3.9 | 12.6 | 25.9 | 3.4 | 16.8 | 14.5 | 3.7 | 11.9 | 33.9 | 15.8 | 25.8 | 30.1 | 16.3 | 26.2 | 11.0 | 4.5 | 8.1 | 23.4 | 7.1 | 17.1 |
| UI-TARS-7B | 58.4 | 12.4 | 36.1 | 50.0 | 9.1 | 32.8 | 20.8 | 9.4 | 18.0 | 63.9 | 31.8 | 50.0 | 63.3 | 20.8 | 53.5 | 30.8 | 16.9 | 24.5 | 47.8 | 16.2 | 35.7 |
| UI-TARS-72B | 63.0 | 17.3 | 40.8 | 57.1 | 15.4 | 39.6 | 18.8 | 12.5 | 17.2 | 64.6 | 20.9 | 45.7 | 63.3 | 26.4 | 54.8 | 42.1 | 15.7 | 30.1 | 50.9 | 17.5 | 38.1 |
| OS-Atlas-4B | 7.1 | 0.0 | 3.7 | 3.0 | 1.4 | 2.3 | 2.0 | 0.0 | 1.5 | 9.0 | 5.5 | 7.5 | 5.1 | 3.8 | 4.4 | 5.6 | 0.0 | 3.1 | 5.0 | 1.7 | 3.7 |
| *+ DiMo-GUI* | 13.6 | 1.4 | 7.7 | 9.6 | 2.8 | 6.7 | 4.1 | 4.7 | 4.2 | 30.6 | 4.5 | 19.3 | 24.3 | 15.1 | 22.2 | 7.5 | 2.2 | 5.1 | 14.6 | 4.0 | 10.6 |
| *+ GMS (w/ Gemini-2.0-Flash)* | **44.2** | **16.6** | **30.8** | **49.0** | **16.1** | **35.2** | **27.9** | **12.5** | **24.1** | **56.3** | **25.5** | **42.9** | **57.6** | **39.6** | **53.5** | **36.4** | **20.2** | **29.1** | **45.2** | **20.2** | **35.7** |
| Δ | 37.1 | 16.6 | 27.1 | 46.0 | 14.7 | 32.9 | 25.9 | 12.5 | 22.6 | 47.3 | 20.0 | 35.4 | 52.5 | 35.8 | 49.1 | 30.8 | 20.2 | 26.0 | 40.2 | 18.5 | 32.0 |
| *+ GMS (w/ Gemini-2.5-Flash-Lite)* | 39.0 | 18.6 | 29.1 | 48.5 | 14.7 | 34.3 | 21.3 | 12.5 | 19.2 | 45.8 | 20.0 | 34.6 | 55.4 | 24.5 | 48.3 | 35.5 | 19.1 | 28.1 | 40.9 | 17.9 | 32.1 |
| Δ | 31.9 | 18.6 | 25.4 | 45.5 | 13.3 | 32.1 | 19.3 | 12.5 | 17.7 | 36.8 | 14.5 | 27.1 | 50.3 | 20.7 | 43.9 | 29.9 | 19.1 | 25.0 | 35.9 | 16.2 | 28.4 |
| UGround-7B | 26.6 | 2.1 | 14.7 | 27.3 | 2.8 | 17.0 | 14.2 | 1.6 | 11.1 | 31.9 | 2.7 | 19.3 | 31.6 | 11.3 | 27.9 | 17.8 | 0.0 | 9.7 | 25.0 | 2.8 | 16.5 |
| *+ DiMo-GUI* | 44.2 | 6.2 | 25.8 | 39.9 | 7.7 | 26.4 | 17.3 | 3.1 | 13.8 | 50.7 | 8.2 | 32.3 | 46.9 | 15.1 | 39.6 | 32.7 | 10.1 | 22.4 | 38.1 | 7.9 | 26.6 |
| *+ GMS (w/ Qwen2.5-VL-7B)* | 44.2 | 13.8 | 29.4 | 56.1 | 15.4 | 39.0 | 33.5 | 17.2 | 29.5 | 54.2 | 25.5 | 41.7 | 59.3 | 34.0 | 53.5 | 37.4 | 18.0 | 28.6 | 47.9 | 19.0 | 36.9 |
| Δ | 15.6 | 11.7 | 14.7 | 28.8 | 12.6 | 22.0 | 19.3 | 15.6 | 18.4 | 22.3 | 22.8 | 22.4 | 27.7 | 22.7 | 25.6 | 19.6 | 18.0 | 18.9 | 22.9 | 16.2 | 20.4 |
| *+ GMS (w/ Gemini-2.0-Flash)* | **60.4** | **24.1** | **42.8** | **63.1** | **24.5** | **46.9** | **35.5** | 14.1 | **30.3** | **62.5** | **27.3** | **47.2** | **71.8** | **43.4** | **65.2** | **52.3** | **27.0** | **40.8** | **57.4** | **25.8** | **45.4** |
| Δ | 33.8 | 22.0 | 28.1 | 35.8 | 21.7 | 29.9 | 21.3 | 12.5 | 19.2 | 30.6 | 24.6 | 27.9 | 40.2 | 32.1 | 37.3 | 34.5 | 27.0 | 31.1 | 32.4 | 23.0 | 28.9 |
| *+ GMS (w/ Gemini-2.5-Flash-Lite)* | 44.8 | 18.6 | 32.1 | 59.6 | 21.7 | 43.7 | 29.9 | **18.8** | 27.2 | 50.0 | **28.2** | 40.6 | 70.1 | 35.8 | 62.2 | 44.9 | 20.2 | 33.7 | 50.2 | 22.8 | 39.7 |
| Δ | 18.2 | 16.5 | 17.4 | 32.3 | 18.9 | 26.7 | 15.7 | 17.2 | 16.1 | 18.1 | 25.5 | 21.3 | 38.5 | 24.5 | 34.3 | 27.1 | 20.2 | 24.0 | 25.2 | 20.0 | 23.2 |
| UGround-V1-7B | 51.9 | 3.4 | 28.4 | 48.0 | 9.1 | 31.7 | 20.0 | 1.6 | 15.3 | 57.6 | 16.4 | 39.8 | 61.6 | 13.2 | 50.4 | 37.4 | 7.9 | 25.0 | 45.6 | 8.4 | 31.4 |
| *+ DiMo-GUI* | 57.8 | 21.4 | 40.1 | 60.1 | 18.1 | 42.5 | 45.7 | 18.8 | 39.1 | **75.7** | 28.2 | 55.1 | **79.7** | 37.7 | 70.0 | 51.4 | **30.3** | 41.8 | 61.7 | 24.3 | 47.4 |
| *+ GMS (w/ Qwen2.5-VL-7B)* | 53.2 | 20.7 | 37.5 | 57.1 | 19.6 | 41.3 | 59.4 | **29.7** | 52.1 | 62.5 | 34.5 | 50.4 | 67.8 | 35.8 | 60.4 | 45.8 | 15.7 | 32.1 | 58.4 | 24.5 | 45.5 |
| Δ | 1.3 | 17.3 | 9.1 | 9.1 | 10.5 | 9.6 | 39.4 | 28.1 | 36.8 | 4.9 | 18.1 | 10.6 | 6.2 | 22.6 | 10.0 | 8.4 | 7.8 | 7.1 | 12.8 | 16.1 | 14.1 |
| *+ GMS (w/ Gemini-2.0-Flash)* | **69.5** | **35.9** | **53.2** | 67.7 | 26.6 | 50.4 | 67.0 | 28.1 | **57.5** | 70.1 | 38.2 | **56.3** | 76.8 | **50.9** | **70.9** | 51.4 | 21.3 | 37.8 | 68.1 | **32.5** | 54.5 |
| Δ | 17.6 | 32.5 | 24.8 | 19.7 | 17.5 | 18.7 | 47.0 | 26.5 | 42.2 | 12.5 | 21.8 | 16.5 | 15.2 | 37.7 | 20.5 | 14.0 | 13.4 | 12.8 | 22.5 | 24.1 | 23.1 |
| *+ GMS (w/ Gemini-2.5-Flash-Lite)* | 59.1 | 29.7 | 44.8 | **72.2** | **30.8** | **54.8** | **70.6** | 17.2 | **57.5** | 69.4 | **38.2** | **55.9** | **78.0** | 45.3 | 70.4 | **57.9** | 29.2 | **44.9** | **68.9** | 31.5 | **54.6** |
| Δ | 7.2 | 26.3 | 16.4 | 24.2 | 21.7 | 23.1 | 50.6 | 15.6 | 42.2 | 11.8 | 21.8 | 16.1 | 16.4 | 32.1 | 20.0 | 20.5 | 21.3 | 19.9 | 23.3 | 23.1 | 23.2 |

Table 1: Main experimental results on the ScreenSpot-Pro dataset. The table reports performance under the proposed GMS framework with different combinations of 'Scanner' and 'Locator' agents, compared against a range of baseline methods. The best accuracy for each setting is highlighted in **bold**, and the second-best is underlined. Relative improvements (in percentage points) are annotated.

# 4 EXPERIMENTS SETUP

## 4.1 DATASET

We evaluate our framework on the **ScreenSpot-Pro** benchmark, which consists of over 1,500 high-resolution desktop screenshots spanning six GUI grounding tasks (Li et al., 2025).

## 4.2 VISION LANGUAGE MODELS

We instantiate our framework with two vision–language models in complementary roles: a general-purpose 'Scanner' for broad visual understanding and instruction following, and a GUI-specialized 'Locator' for precise element localization.

To demonstrate that the framework effectively exploits each model's strengths, we select two well-known grounding-focused model families, each with fewer than 7B parameters: OS-Atlas (Wu et al., 2024) and UGround (Gou et al., 2025).

For the 'Scanner' role, we balance cost and model availability (including both open-weight and closed-source models) and choose: Qwen2.5-VL (Bai et al., 2025) and Gemini (Google, 2025; Comanici et al., 2025; Team et al., 2024; 2025b).

## 4.3 METRICS

We use **accuracy** as the evaluation metric. Formally, let $\hat{\mathbf{p}} = (x, y)$ denote the predicted coordinate and $\mathcal{B} = [x_{\min}, x_{\max}] \times [y_{\min}, y_{\max}]$ denote the ground-truth bounding box. We define an indicator function:

$$\mathbb{I}(\hat{\mathbf{p}} \in \mathcal{B}) = \begin{cases} 1, & \text{if } x_{\min} \leq x \leq x_{\max} \text{ and } y_{\min} \leq y \leq y_{\max}, \\ 0, & \text{otherwise.} \end{cases}$$

Then, the accuracy over $N$ samples is: $Accuracy = \frac{1}{N} \sum_{i=1}^{N} \mathbb{I}(\hat{\mathbf{p}}_i \in \mathcal{B}_i)$.

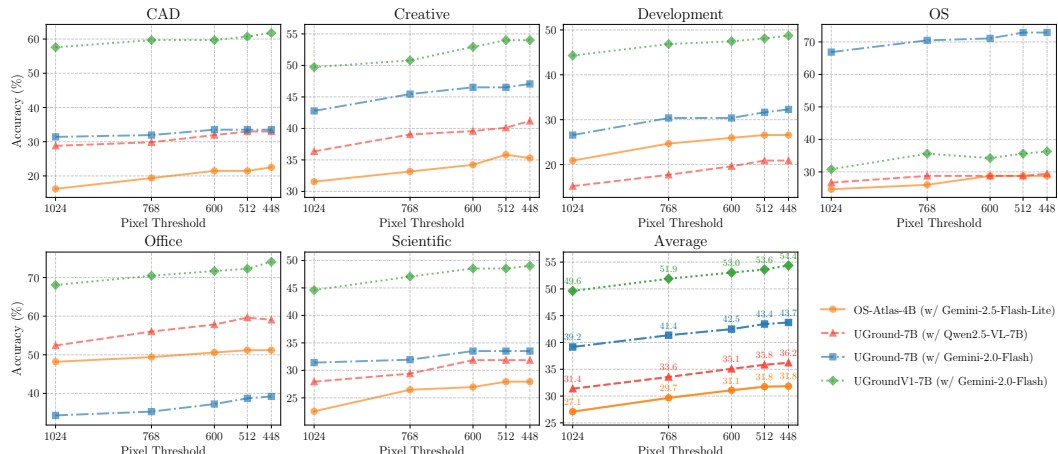

Figure 4: Experimental results illustrating the impact of decreasing the pixel number threshold from 1024 to 448 in the hierarchical search constraints. The figure reports the accuracy of six sub-categories and the overall accuracy across four agentic combinations.

## 4.4 IMPLEMENTATION DETAILS

We obtain all open-weight models from their official repositories on HuggingFace. For these fine-tuned grounding models, we set the temperature to $0.0$ to ensure faithfulness. For closed-source models, we perform inference via the OpenRouter platform. To ensure consistency and efficiency, we adopt the default inference settings: temperature $= 0.7$ and $\text{top}_p = 0.95$. All experiments were conducted on a machine equipped with two NVIDIA A100 80GB GPUs and 1,000 GB of RAM. The prompts and baseline introduction are provided in Appendix G and Appendix E, respectively.

## 5 EXPERIMENT RESULTS

We evaluate our proposed framework on the ScreenSpot-Pro benchmark, with results presented in Table 1. Our framework consistently outperforms all baselines across multiple settings, including strong fine-tuned models (up to 72B parameters) and DiMo-GUI. The improvements are particularly substantial across various sub-categories, covering both text and icon grounding tasks, with relative gains ranging from $100\%$ to over $1000\%$. We highlight the following key findings:

**Effectiveness in Low-Performance Settings.** The OS-Atlas-4B model performs poorly under direct inference, achieving only $13\%$ accuracy even with DiMo-GUI. Remarkably, when integrated into our framework and paired with Gemini models, each of which yields less than $3\%$ accuracy individually, the combined system achieves $30\%$ accuracy. This represents a $10\times$ improvement over the original results and a $2\times$ improvement over DiMo-GUI. These findings highlight the framework's ability to coordinate weaker models into a highly effective cooperative system by assigning them specialized roles.

**Superior Performance on Icon Grounding Tasks.** Existing methods, including DiMo-GUI and large fine-tuned models, typically underperform on icon-related grounding tasks compared to text-based tasks. In contrast, our GMS framework substantially alleviates this disparity. By leveraging the synergy between the generalist 'Scanner' and specialist 'Locator' modules, our method boosts icon grounding accuracy from $1.7\%$ to $20\%$ on OS-Atlas-4B (a $1076\%$ improvement) and from $2.8\%$ to $25.8\%$ on UGround-7B (an $821\%$ improvement).

**Scalability and Model Flexibility.** Our framework demonstrates strong scalability. Even with relatively small 'Scanner' models, such as 7B or flash-Gemini variants, the performance gains remain substantial. Furthermore, the results indicate that stronger general models lead to better outcomes. Given the framework's flexible and modular design, integrating more capable models (e.g., stronger Gemini variants) may yield further performance improvements.

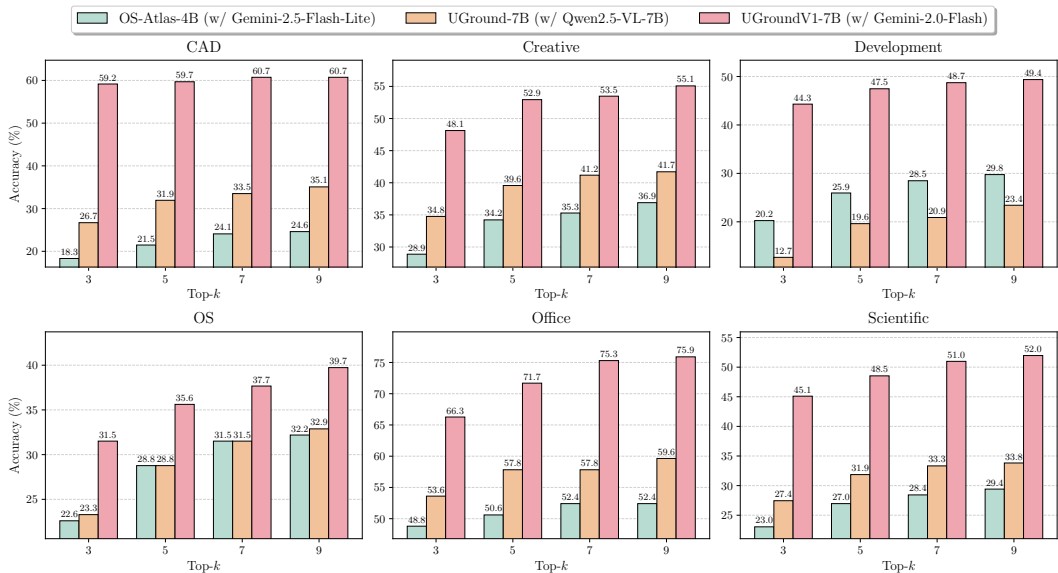

Figure 5: Experimental results showing the impact of increasing the top-$k$ selection value from 3 to 9. The figure reports the accuracy across six sub-categories under four agentic combinations.

## 6 TEST-TIME SCALING

Our framework integrates the concept of test-time scaling into the hierarchical search process. The inference time can be flexibly controlled by adjusting two key factors: the top-$k$ selection parameter at each search step and the pixel threshold used during the process. In this section, we analyze the impact of these parameters on the 15 most challenging subsets.

### 6.1 IMPACT OF PIXEL VALUE THRESHOLD

We begin by evaluating the impact of different pixel thresholds, with accuracy results shown in Figure 4. As the threshold decreases from 1024 to 512, which allows the 'Scanner' agents to capture finer-grained details of the GUI interface, we observe a consistent increase in accuracy. Notably, even at the higher threshold of 1024, the framework maintains strong performance, with only a marginal drop in accuracy compared to lower thresholds. This result highlights the robustness and effectiveness of our proposed framework. Moreover, the improvements observed with decreasing thresholds suggest promising test-time scaling capabilities. Given that many original images in the ScreenSpot-Pro dataset exceed 3000 pixels in resolution, these findings indicate that the framework is well suited for high-resolution settings, which are critical in GUI understanding tasks.

### 6.2 IMPACT OF TOP-K REGION SELECTION

We further examine the impact of the $k$-value in the top-$k$ region selection mechanism, which guides the deeper stages of hierarchical search. As shown in Figure 5, increasing $k$ from 3 to 9 leads to a gradual improvement in accuracy. This trend aligns with intuition, as a larger search space allows the 'Scanner' agent to explore more potentially relevant subregions. Since in high-resolution settings, where input images are especially large, the agent may fail to cover all important areas. A smaller $k$ can lead to the omission of critical regions, particularly when the agent assigns low confidence to relevant areas due to limited context or weaker understanding.

The most significant performance gain occurs when increasing $k$ from 3 to 5, suggesting that the 'Scanner' agent possesses a baseline level of capability, and a modest expansion of the search space greatly enhances its effectiveness. Beyond this point, further improvements are observed, but with diminishing returns. We also find that the choice of 'Scanner' agent influences the model's sensitivity to changes in $k$. For example, the Qwen2.5-VL-7B model shows the most pronounced improvement as $k$ increases; on the *Development* split, accuracy rises from 12.7% to 23.4%. In contrast, when

| Scanner Agent | Locator Agent | Inference Method | Overall Accuracy (%) |
|---|---|---|---|
| Gemini-2.0-Flash | OS-Atlas-4B | GMS | 35.67 |
| | | *w/o Cross-Modal Verification* | 27.83($\downarrow$ 7.84) |
| | | *w/o Multi-Agent Consensus* | 33.05($\downarrow$ 2.62) |
| | | *w/o Adaptive Resolution Enhancement* | 31.59($\downarrow$ 4.08) |
| Gemini-2.5-Flash-Lite | UGround-7B | GMS | 39.72 |
| | | *w/o Cross-Modal Verification* | 33.57($\downarrow$ 6.15) |
| | | *w/o Multi-Agent Consensus* | 37.20($\downarrow$ 2.52) |
| | | *w/o Adaptive Resolution Enhancement* | 34.91($\downarrow$ 4.81) |

Table 2: Ablation results after individually removing each of the three crucial stages from our framework. Overall accuracy drops significantly compared to the full framework, highlighting the cooperative nature and effectiveness of the proposed architecture.

stronger Gemini models are used as the 'Scanner' agent, the benefit of increasing $k$ is less substantial. This aligns with expectations, as stronger vision-language models are generally more confident and accurate in identifying the correct region, thereby reducing the need for a large search space.

## 7 ABLATION STUDY

To validate the effectiveness of each key component in our proposed framework, we conduct a series of ablation studies using three modified baselines:

- Deletion of Cross-Modal Verification: The 'Locator' agent's predicted coordinates are passed directly, without any confidence-based filtering.
- Deletion of Multi-Agent Consensus: The 'Locator' agent always selects the coordinate with the highest confidence score, bypassing the consensus selection mechanism.
- Deletion of Adaptive Resolution Enhancement: The 'Locator' and 'Scanner' agents no longer collaborate; the output of the 'Locator' alone is used as the final click instruction.

The experimental results, presented in Table 2, show that removing any component leads to noticeable performance degradation. Among these experiments, the removal of cross-modal verification has the most severe impact. This is likely due to the absence of confidence filtering, which causes the 'Scanner' agent to receive multiple candidate regions without sufficient guidance, making effective reasoning difficult, especially under long-context constraints or limited output reasoning capacity.

In contrast, the removal of multi-agent consensus has the least impact on performance. This finding reflects the strong evaluation capabilities of the 'Scanner' agent, which can reliably assess each candidate to determine whether it contains the target position. As a result, selecting only the candidate with the highest verification score still yields strong performance compared to the other two conditions. This further supports the design intuition of assigning distinct and complementary responsibilities to different agent types within our framework.

## 8 CONCLUSION

We propose **GMS**: **G**eneralist Scanner **M**eets **S**pecialist Locator, a synergistic coarse-to-fine framework that employs hierarchical search with test-time scaling. Drawing inspiration from how humans approach GUI grounding tasks, GMS introduces two specialized roles: the 'Scanner' and the 'Locator'. This division enables cooperative behavior, allowing each agent to focus on the subtask that aligns with its respective strengths. The 'Scanner' performs coarse region localization, while the 'Locator' is responsible for precise coordinate prediction within the identified region. Extensive experiments on the ScreenSpot-Pro dataset demonstrate the effectiveness of GMS. The framework not only surpasses strong baselines but also substantially boosts the performance of two relatively weak models when combined, achieving nearly double their original accuracy without any additional fine-tuning. Ablation studies further confirm the robustness of the framework, showing that each stage contributes significantly to overall performance. These results underscore the practical applicability of GMS to real-world GUI grounding scenarios and highlight its potential as a general paradigm for agent collaboration in vision–language tasks.

ETHICS STATEMENT

Ethical considerations play a central role in this research. All models used in this study are either open-weight or widely adopted within the scientific community, ensuring transparency and reproducibility. The proposed GMS framework aims to advance the capabilities of current VLM agents for the GUI grounding task, contributing to real-world applications without introducing or reinforcing harmful biases. No personally identifiable information or sensitive data is involved in this work. We are committed to responsible research practices and advocate for the transparent reporting and ethical deployment of AI technologies in ways that serve the broader interests of society.

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

## A    THE USE OF LARGE LANGUAGE MODELS (LLMS)

We used LLMs to assist with the phrasing and grammar of the manuscript. The LLMs were used strictly as a writing aid and did not contribute to the scientific ideation, methodology, or results presented in this paper.

## B    IMPLEMENTATION DETAILS

For the closed-source models, we perform inference using the OpenRouter platform, and all use the default provider. All experiments were conducted between August 15 and September 15 on a machine equipped with two NVIDIA A100 80GB GPUs and 1000GB of RAM.

### B.1    BENCHMARK DATASET DISCUSSION

The **ScreenSpot-Pro** benchmark dataset poses challenges such as diverse icons, layouts, and application-specific styles, as well as large input sizes and heterogeneous content, making it a suitable resource for evaluating model robustness.

### B.2    MAIN EXPERIMENT DETAILS

In the main experimental results shown in Table 1, we compare our approach with several baselines: (1) direct inference using the original models and (2) DiMo-GUI, one of the strongest existing baselines, which incorporates the concept of test-time scaling. To balance performance and computational efficiency, we set the threshold for hierarchical image search to 600 pixels, meaning that the search terminates when either the image width or height falls below this threshold.

### B.3    EXPERIMENTAL MODEL INTRODUCTION

Here we briefly introduce the four types of models that are used in the main experiments, as the general 'Scanner' and the specialist 'Locator'.

#### B.3.1    'LOCATOR' MODELS

- OS-Atlas (Wu et al., 2024): An open-source foundational action model series for GUI agents, trained on 2.3 million cross-platform screenshots and 13 million UI elements.
- UGround (Gou et al., 2025): A universal visual-only grounding model family that predicts pixel-level element coordinates using only visual input, trained on 1.3 million screenshots containing 10 million GUI elements.

#### B.3.2    'SCANNER' MODELS

- Qwen2.5-VL (Bai et al., 2025): A recent multimodal vision–language model series, available in multiple sizes, that offers strong visual understanding.
- Gemini (Google, 2025; Comanici et al., 2025; Team et al., 2024; 2025b): Google's family of multimodal models, capable of processing text, images, audio, and code, and designed for broad AI applications including chat and search.

## C    FURTHER RELATED WORK

Here, we further discuss related work concerning the use of test-time scaling in the field of GUI grounding:

### C.1    TEST-TIME SCALING

Test-time scaling refers to techniques that improve model performance at inference without modifying model parameters, typically by increasing computation or using additional resources (Muennighoff et al., 2025; Snell et al., 2024; Zhang et al., 2025c). In GUI grounding, test-time scaling has been

used to improve localization through action histories, external knowledge retrieval, zoom-in searches, and adaptive focus refinement (Wu et al., 2025a; Nguyen, 2025; Nakano et al., 2022). These methods aim for greater accuracy via extended reasoning and iterative attention.

# D  ADDITIONAL BASELINE PERFORMANCE

Due to the page limitations in the main paper, we also report the raw performance of several additional vision-language models on the test benchmark, which we list in Table 3.

| Base Model | Development | | | Creative | | | CAD | | | Scientific | | | Office | | | OS | | | Average | | |
|---|---|---|---|---|---|---|---|---|---|---|---|---|---|---|---|---|---|---|---|---|---|
| | text | icon | avg | text | icon | avg | text | icon | avg | text | icon | avg | text | icon | avg | text | icon | avg | text | icon | avg |
| Gemma-3-27B | 0.0 | 0.0 | 0.0 | 2.0 | 0.0 | 1.2 | 1.0 | 1.6 | 1.1 | 3.5 | 0.0 | 2.0 | 1.1 | 0.0 | 0.9 | 0.0 | 0.0 | 0.0 | 1.3 | 0.2 | 0.9 |
| Phi-4-Multimodal | 0.0 | 0.7 | 0.3 | 1.5 | 0.0 | 0.9 | 0.5 | 1.6 | 0.8 | 2.1 | 0.0 | 1.2 | 1.7 | 5.7 | 2.6 | 0.0 | 0.0 | 0.0 | 1.0 | 0.8 | 0.9 |
| SeeClick | 0.6 | 0.0 | 0.3 | 1.0 | 0.0 | 0.6 | 2.5 | 0.0 | 1.9 | 3.5 | 0.0 | 2.0 | 1.1 | 0.0 | 0.5 | 2.8 | 0.0 | 1.5 | 1.8 | 0.0 | 1.1 |
| Claude Sonnet 4 | 1.3 | 3.4 | 2.3 | 1.5 | 0.7 | 1.2 | 2.5 | 0.0 | 1.9 | 1.4 | 1.8 | 1.6 | 0.6 | 1.9 | 0.9 | 0.0 | 0.0 | 0.0 | 1.3 | 1.5 | 1.4 |
| Qwen2-VL-7B | 2.6 | 0.0 | 1.3 | 1.5 | 0.0 | 0.9 | 0.5 | 0.0 | 0.4 | 6.3 | 0.0 | 3.5 | 3.4 | 1.9 | 3.0 | 0.9 | 0.0 | 0.5 | 2.5 | 0.2 | 1.6 |
| GLM-4.5V | 0.0 | 1.4 | 0.7 | 1.5 | 2.1 | 1.8 | 5.6 | 0.0 | 4.2 | 2.8 | 1.0 | 2.0 | 1.7 | 1.9 | 1.7 | 0.0 | 0.0 | 0.0 | 2.1 | 1.2 | 1.8 |
| GPT-5 | 2.6 | 0.7 | 1.7 | 4.5 | 4.2 | 4.4 | 7.6 | 7.8 | 7.7 | 4.2 | 1.8 | 3.1 | 4.5 | 3.8 | 4.3 | 0.0 | 0.0 | 0.0 | 4.3 | 2.6 | 3.7 |
| Gemini-2.5-Pro | 4.5 | 2.8 | 3.7 | 7.6 | 5.6 | 6.7 | 14.2 | 1.6 | 11.1 | 4.9 | 6.4 | 5.5 | 7.3 | 3.8 | 6.5 | 2.8 | 1.1 | 2.0 | 7.5 | 3.8 | 6.1 |
| ShowUI-2B | 16.9 | 1.4 | 9.4 | 9.1 | 0.0 | 5.3 | 2.5 | 0.0 | 1.9 | 13.2 | 7.3 | 10.6 | 15.3 | 7.5 | 13.5 | 10.3 | 2.2 | 6.6 | 10.8 | 2.6 | 7.7 |

Table 3: Additional baseline results of various vision-language models on the GUI grounding benchmark. Despite their strong general capabilities, these models perform poorly on this specific task.

From the results presented in the table, it is evident that most state-of-the-art vision-language models, including those from families such as GPT-5 and Gemini-2.5-Pro, perform poorly on the GUI grounding benchmark despite their strong general vision capabilities. This highlights a critical limitation in their ability to handle fine-grained, domain-specific grounding tasks. Nevertheless, their robust visual perception suggests that they can still serve effectively as visual front-ends to parse and understand GUI images. These findings underscore the importance of fully leveraging the intrinsic capabilities of such models, rather than relying solely on scaling up data or fine-tuning larger parameter models.

# E  BASELINE INTRODUCTION

We compare our GMS framework with several baselines, as shown in Table 1 and Table 3. Below, we introduce each baseline to provide clarification.

- *GPT-4o* (OpenAI et al., 2024): OpenAI's flagship multimodal model that seamlessly understands and generates text, images, and audio. It enables faster, more natural real-time interactions while maintaining strong reasoning and accuracy.

- *Gemma3-27B* (Team et al., 2025c): Google's 27B-parameter version of their Gemma 3 model family. It's a high-capacity, multimodal model that accepts both text and image inputs, supports an expanded 128K context window, works across 140 languages.

- *Phi-4-Multimodal* (Abdin et al., 2024): Microsoft's 5.6B-parameter model that can process text, vision, and speech (audio) inputs in a unified system. It supports a long 128K token context, uses a "mixture of LoRAs" approach for modality-adapters.

- *SeeClick* (Cheng et al., 2024): A visual GUI agent that automates tasks like clicking or typing by observing only interface screenshots, without needing structured representations such as HTML or accessibility trees.

- *Claude-Sonnet-4* Anthropic (2025): A mid-tier model in Anthropic's Claude 4 family, designed to balance strong reasoning and coding ability with efficiency and accessibility.

- *Qwen2-VL-7B* (Wang et al., 2024b; Bai et al., 2023): A 7B-parameter vision-language model from Alibaba's Qwen2-VL family.

- *GLM-4.5V* (Team et al., 2025a): ZhipuAI's flagship vision-language model built on GLM-4.5-Air, activating 12B of its 106B parameters per pass to balance efficiency with strong multimodal reasoning.

- *Gemini-2.0-Flash* (Google, 2025): Google's high-performance, multimodal model in the Gemini 2.0 family designed for the "agentic era".
- *Gemini-2.5-Flash-Lite* (Comanici et al., 2025): Google's cost- and latency-optimized variant in the Gemini 2.5 model series, designed for high-volume, real-world use.
- *GPT-5* (OpenAI, 2025): OpenAI's next-generation multimodal model that advances beyond GPT-4o with stronger reasoning, longer context handling, and more efficient real-time interaction across text, vision, and audio.
- *Gemini-2.5-Pro* (Comanici et al., 2025): Google's top-tier reasoning model in the Gemini 2.5 family, designed to tackle complex problems across modalities, including text, audio, images, video, and even whole code repositories.
- *ShowUI-2B* (Lin et al., 2024): A lightweight vision-language-action model from ShowLab, built for GUI agents to understand and interact with graphical user interfaces via screenshots.
- *CogAgent-18B* (Hong et al., 2024): An open-source vision-language model (VLM) developed by THUDM and Zhipu AI, specifically optimized for understanding and interacting with graphical user interfaces (GUIs).
- *Aria-UI* (Yang et al., 2025): A multimodal model for GUI grounding that maps language instructions to specific interface elements using only vision (screenshots), foregoing HTML or accessibility trees (AXTrees) as auxiliary input.
- *Claude (Computer Use)* (Hu et al., 2024): A GUI-agent extension of Claude 3.5 Sonnet that enables the model to observe screenshots of a user's computer and issue desktop actions (mouse, keyboard, clicks) to automate tasks.
- *UI-TARS-7B* (Qin et al., 2025): A 7B-parameter vision-language model from ByteDance designed for native GUI automation, capable of controlling both web and desktop applications via only screenshot input.
- *UI-TARS-72B* (Qin et al., 2025): 72B-parameter version of UI-TARS.

For the baselines not presented in the previous paper, we conduct the evaluations ourselves, with the inference prompts provided in Appendix G. For the baselines included in the previous paper, we directly use the results reported by Wu et al. (2025a), which also correspond to the ScreenSpot-Pro leaderboard data. Regarding the general-purpose vision-language models, we select recent and strong models to demonstrate their raw performance on the direct GUI grounding task, which turns out to be rather poor.

## F  DISCUSSION

In this section, we elaborate on the proposed framework and present further analysis of the associated experiments, demonstrating its superior performance and the novel insights it yields relative to existing methods.

### F.1  COGNITION AND ARCHITECTURAL INSIGHTS

Here, we discuss several key aspects in which our GMS framework departs from prior works, highlighting the unique insights underlying our design:

(1) Cognitive-inspired task decomposition based on the dual-stream model, separating semantic attention from motor-level localization.

(2) Hierarchical attention and cross-modal verification that iteratively refine the search space, replacing brittle single-pass grounding.

(3) Asymmetric multi-agent collaboration, with a generalist scanner for abstraction and a specialist locator for spatial precision.

(4) Late-stage fusion through multi-resolution decision making, aligning global predictions with fine-grained local cues.

These insights allow GMS to achieve a stronger balance between global semantic reasoning and local spatial precision than prior approaches.

## F.2 On the Possibility of Comparing with Additional Baselines

In this paper, we primarily compare our framework with DiMo-GUI, one of the strongest existing baselines on the GUI grounding benchmark. Although numerous related works report results on this benchmark, their performance under comparable grounding model settings consistently falls short of DiMo-GUI. Therefore, we focus our comparison on DiMo-GUI to balance both reproducibility costs and page constraints. Given that our framework outperforms DiMo-GUI, it is reasonable to infer that it also surpasses other baselines that perform worse than DiMo-GUI.

## G  Inference Prompts

We present the manually designed inference prompts employed in the experiments shown in Figures 6 through 18.

The first thing we want to note is that, for the specific fine-tuned grounding models, we use the same inference prompt as the researchers in the previous work, without making any modifications. The inference-related code is written based on the HuggingFace repository example code, including some resizing and transformations for certain models such as OS-Atlas-4B.

The second point to note is that each main prompt designed for our GMS framework has two versions, depending on the names of certain subsets. Subsets such as "ppt_windows" or "word_macos" clearly indicate the application name (here "powerpoint" and "word"). However, there are three special subsets, namely "common_linux", "common_windows", and "common_macos", which do not contain specific application names. For this reason, we provide two versions of each prompt: the first is used for most subsets, while the second is used for the three special subsets mentioned here.

---

**Hierarchical Attention Allocation Prompt (Initial Level & Normal Version)**

I have provided you a screenshot of my desktop containing the interface of the {application_name} application running on the {system_name} system. Where should I click if I want to DIRECTLY perform the following operation in the {application_name}: **{instruction}**? Provide the possibilities for each region (Region 1 to Region 9, ordered from left to right, top to bottom) with a score between 0 and 100. Your output MUST follow this format: "Region X: SCORE (explanation)".

---

Figure 6: The hierarchical attention allocation prompt for the initial level (search depth = 0) and the normal subsets.

---

**Hierarchical Attention Allocation Prompt (Initial Level & Special Version)**

I have provided you a screenshot of my desktop using {system_name} system. Where should I click if I want to DIRECTLY perform the following operation in the {application_name}: **{instruction}**? Provide the possibilities for each region (Region 1 to Region 9, ordered from left to right, top to bottom) with a score between 0 and 100. Your output MUST follow this format: "Region X: SCORE (explanation)".

---

Figure 7: The hierarchical attention allocation prompt for the initial level (search depth = 0) and the special subsets.

---

**Hierarchical Attention Allocation Prompt (Non-Initial Level & Normal Version)**

Where should I click if I want to DIRECTLY perform the following operation in the {application_name}: **{instruction}**? Provide the possibilities for each region (Region 1 to Region 9, ordered from left to right, top to bottom) with a score between 0 and 100. Your output MUST follow this format: "Region X: SCORE (explanation)".

---

Figure 8: The hierarchical attention allocation prompt for the non-initial level (search depth $\geq 1$) and the normal subsets.

---

**Hierarchical Attention Allocation Prompt (Non-Initial Level & Special Version)**

Where should I click if I want to DIRECTLY perform the following operation: **{instruction}**? Provide the possibilities for each region (Region 1 to Region 9, ordered from left to right, top to bottom) with a score between 0 and 100. Your output MUST follow this format: "Region X: SCORE (explanation)".

---

Figure 9: The hierarchical attention allocation prompt for the non-initial level (search depth $\geq 1$) and the special subsets.

---

**Scanner Region Verification Prompt (Normal Version)**

You need to check if the image region from my desktop screenshot contains the button or icon for me to DIRECTLY perform the following operation in the {application}: **{instruction}**. You are required to output your reasoning process first, and then provide your final answer in the format: <answer>yes/no</answer>.

---

Figure 10: The region verification prompt that instructed the 'Scanner' agent to filter the region of interest (for normal subsets).

---

**Scanner Region Verification Prompt (Special Version)**

You need to check if the image region from my desktop screenshot contains the button or icon for me to DIRECTLY perform the following operation: **{instruction}**. You are required to output your reasoning process first, and then provide your final answer in the format: <answer>yes/no</answer>.

---

Figure 11: The region verification prompt that instructed the 'Scanner' agent to filter the region of interest (for special subsets).

---

**Cross-Modal Verification Prompt (Normal Version)**

I want to DIRECTLY perform the following operation in the {application}: **{instruction}**. First, I'm showing you the original screenshot image, followed by a cropped region from it. You need to determine whether this cropped region is likely to contain the target button, area, or icon for the operation. Answer with `<relevance>yes/no</relevance>` and provide your reasoning within `<reasoning>...</reasoning>`.
This is the original screenshot image: `<Image1>`
And this is the cropped region from the original screenshot image: `<Image2>`

Figure 12: The cross-modal verification prompt that instructed the 'Scanner' agent to verify the cropped region (for normal subsets).

---

**Cross-Modal Verification Prompt (Special Version)**

I want to DIRECTLY perform the following operation: **{instruction}**. First, I'm showing you the original screenshot image, followed by a cropped region from it. You need to determine whether this cropped region is likely to contain the target button, area, or icon for the operation. Answer with `<relevance>yes/no</relevance>` and provide your reasoning within `<reasoning>...</reasoning>`.
This is the original screenshot image: `<Image1>`
And this is the cropped region from the original screenshot image: `<Image2>`

Figure 13: The cross-modal verification prompt that instructed the 'Scanner' agent to verify the cropped region (for special subsets).

---

**Scanner Adaptive Resolution Enhancement Prompt (Normal Version)**

I want to DIRECTLY perform this operation in the {application} on my desktop: **{instruction}**.
I have extracted a candidate region and divided it into 5x5 smaller regions (numbered 1 to 25 from left to right, top to bottom). Please identify which of the 25 regions is the most relevant (return only one region you are most confident about). Then, within that region, tell me which of the 9 inner zones the target click point is closest to. (Choose from: top left, top center, top right, center left, center, center right, bottom left, bottom center, bottom right)
First, provide your reasoning process, and then return your final answer in the following format:
`<index>xxx</index>`
`<location>xxx</location>`

Figure 14: The adaptive resolution prompt that instructed the 'Scanner' agent to identify a possible fine-grained region (for normal subsets).

---

**Scanner Adaptive Resolution Enhancement Prompt (Special Version)**

I want to DIRECTLY perform this operation on my desktop: **{instruction}**.
I have extracted a candidate region and divided it into 5x5 smaller regions (numbered 1 to 25 from left to right, top to bottom). Please identify which of the 25 regions is the most relevant (return only one region you are most confident about). Then, within that region, tell me which of the 9 inner zones the target click point is closest to. (Choose from: top left, top center, top right, center left, center, center right, bottom left, bottom center, bottom right)
First, provide your reasoning process, and then return your final answer in the following format:
```
<index>xxx</index>
<location>xxx</location>
```

Figure 15: The adaptive resolution prompt that instructed the 'Scanner' agent to identify a possible fine-grained region (for special subsets).

---

**OS-Atlas-4B Grounding Prompt**

In the screenshot of this web page, please give me the coordinates of the element I want to click on according to my instructions(with point).\n"{}"

Figure 16: The instruction for the OS-Atlas-4B model to output grounding coordinates.

---

**UGround-7B Grounding Prompt**

In the screenshot, where are the pixel coordinates (x, y) of the element corresponding to \"{}\"?

Figure 17: The instruction for the UGround-7B model to output grounding coordinates.

---

**UGround-V1-7B Grounding Prompt**

Your task is to help the user identify the precise coordinates (x, y) of a specific area/element/object on the screen based on a description.
- Your response should aim to point to the center or a representative point within the described area/element/object as accurately as possible.
- If the description is unclear or ambiguous, infer the most relevant area or element based on its likely context or purpose.
- Your answer should be a single string (x, y) corresponding to the point of the interest.
Description: {instruction}
Answer:

Figure 18: The instruction for the UGround-V1-7B model to output grounding coordinates.

**Baseline Models Grounding Prompt**

I want to DIRECTLY perform this operation in the {application} on my desktop: **{instruction}**. You should provide the target CLICK pixel coordinate (x, y) in the ORIGINAL image. You must output only integer coordinate values. For example: '123, 456' or '(123, 456)'.

Figure 19: The instruction for the baseline models to output grounding coordinates.

