# OpenReview forum: "Generalist Scanner Meets Specialist Locator: A Synergistic Coarse-to-Fine Framework for Robust GUI Grounding"
_ICLR.cc/2026/Conference — ICLR 2026 Conference Withdrawn Submission_

### Official Review · Reviewer_5eGP · 2025-11-01

**Soundness:** 2
**Presentation:** 3
**Contribution:** 1
**Rating:** 2
**Confidence:** 4

**Summary:**

This paper addresses the GUI grounding problem, one of the key challenges for GUI agent models. The authors propose a coarse-to-fine grounding approach that integrates two modules, a scanner and a locator. Experiments are conducted on GUI grounding benchmarks across diverse UI environments, and three different models are employed for the scanner component.

**Strengths:**

- The proposed method is clear and the motivation is intuitive.
- Improvments specially using Gemini as Scanner model seems clear.

**Weaknesses:**

- The major concern lies in the novelty of the work. The coarse-to-fine grounding concept has already been proposed in prior work (R-VLM ACL 2025), yet this paper neither cites nor discusses the differences. The overall idea appears to overlap significantly, except for the use of a different vision-language model (VLM) for the scanner component.

- Considering the performance gap between Qwen2.5-VL and Gemini, the improvement observed when using Gemini as the scanner seems rather trivial and expected, as it primarily reflects the stronger base model rather than methodological innovation.

- The experiments are limited to GUI grounding benchmarks and do not include evaluations on broader GUI agent tasks. It remains unverified whether the proposed approach generalizes to realistic interactive settings such as AITW, Multimodal-Mind2Web, or MiniWob.

**Questions:**

Please see the Weakness.

---

### Official Review · Reviewer_CZan · 2025-11-01

**Soundness:** 3
**Presentation:** 2
**Contribution:** 2
**Rating:** 4
**Confidence:** 4

**Summary:**

This paper introduces GMS, a training-free multi-agent framework that emulates human-like grounding by assigning complementary roles to generalist and specialist models, achieving substantial gains without additional fine-tuning. Extensive experiments demonstrate its effectiveness.

**Strengths:**

1. The experimental evaluation is generally comprehensive.

2. The method section is fairly complete, and the approach yields performance improvements.

**Weaknesses:**

1. The paper is not clearly written. For example, the motivation in the introduction is somewhat confusing and lacks a clear statement of the specific problem being addressed.

2. The novelty is limited. It lacks comparisons with similar test-time scaling approaches, such as [1].

3. Some experiments are still missing. For instance, the method section introduces several components to implement the full agent— including the verification mechanism and multi-agent debate, yet the overhead attributable to each stage (token consumption and time cost) does not appear to be quantified.

[1] Visual Test-time Scaling for GUI Agent Grounding. ICCV 2025.

**Questions:**

1. The method section mentions “verification” multiple times; however, the ablation study does not clearly specify which verification component is being evaluated.

---

### Official Review · Reviewer_E2qj · 2025-11-01

**Soundness:** 2
**Presentation:** 3
**Contribution:** 2
**Rating:** 4
**Confidence:** 5

**Summary:**

This paper proposes GMS (Generalist Scanner Meets Specialist Locator), a training-free, multi-agent framework for grounding natural language queries in graphical user interfaces (GUIs). The method integrates a generalist vision-language model (Scanner) for broad semantic perception with a specialist grounding model (Locator) for fine-grained coordinate prediction in a coarse-to-fine manner. Experiments on the ScreenSpot-Pro benchmark show that while each model performs poorly in isolation, their integration boosts grounding accuracy (e.g., from below 4% to 36% for OS-Atlas-4B).

**Strengths:**

1. The paper presents a straightforward and modular design that separates generalist perception and specialist localization, making the overall idea easy to follow and potentially adaptable to other multimodal grounding tasks.
2. The experiments, though limited in scope, demonstrate that integrating a generalist and a specialist model can lead to performance improvements.

**Weaknesses:**

*Method
1. The contributions are limited. The proposed framework mainly combines two existing large models (i.e., a generalist vision-language model and a specialist grounding model) through iterative refinement. While the design is functional, it feels largely engineering-driven and lacks deeper methodological insight to justify why such a combination leads to substantial improvements.
2. Deploying two large models within a GUI agent pipeline raises significant concerns about cost and inference efficiency. However, the paper does not provide any quantitative analysis of runtime, memory consumption, or scalability. Without such an evaluation, it is difficult to assess the practicality of the proposed approach for real-world GUI interaction systems.

*Experiments

Experiments are conducted only on a single dataset (ScreenSpot-Pro), which limits the generalizability of the conclusions. As there are some related datasets such as ScreenSpot and ScreenSpot-v2, it would be better to include results on these benchmarks to demonstrate the superiority.

In general, I do not think this paper is ready for publication yet, as both the methodology and experiments remain relatively underdeveloped.

**Questions:**

Please refer to the weaknesses.

---

### Official Review · Reviewer_3PJU · 2025-11-01

**Soundness:** 2
**Presentation:** 2
**Contribution:** 1
**Rating:** 0
**Confidence:** 5

**Summary:**

The paper primarily focuses on the GUI Grounding task. The authors propose a coarse-to-fine framework, which includes several modules: Hierarchical attention allocation, Iterative focus refinement, Cross-modal verification, Multi-agent consensus, Adaptive resolution enhancement. The experiments and comparisons were only conducted on the screenspot-pro dataset.

**Strengths:**

The writing is easy to follow.

**Weaknesses:**

1. The proposed method requires multiple rounds of image cropping into smaller sub-regions for the MLLM to perform repeated inference and decision-making for a single Grounding task (i.e., the Scanner module).
  - First, the strategy of cropping the original image based on a preset grid is query-unaware. The MLLM receives only local information and lacks global context, which is likely to impair the decision quality.
  - Second, the time and economic cost associated with these repeated inferences are significant and cannot be ignored, especially since a complex task often requires multiple grounding steps to complete. The proposed method, therefore, appears highly impractical.
2. The proposed method relies on numerous heuristic hyperparameters, such as the $125 \times 125$ pixels (L232-233), the $3 \times 3$ subgrid (L260-261), and the $\times 5$ upscale factor for $C^*$, among others. The paper must include an ablation study to validate the necessity and rationality of these specific choices.
3. Based on the experiments, the Locator agent module selected DiMo-GUI as the grounding model. The paper needs to justify why only this specific model was chosen and whether other potential grounding models could be used instead.
4. The proposed method was only evaluated on a single benchmark (screenspot-pro). It is essential to validate its performance on multiple benchmarks, such as ScreenSpot-v2 and OSWorld, to demonstrate generalizability.

**Questions:**

Please find the question in the Weaknesses section.

---

### Note · Authors · 2025-12-01

I have read and agree with the venue's withdrawal policy on behalf of myself and my co-authors.